# How neocarcerand Octacid4 self-assembles with guests into irreversible noncovalent complexes and what accelerates the assembly

Yuan-Ping Pang [ID] [1✉]

Cram's supramolecular capsule Octacid4 can irreversibly and noncovalently self-assemble with small-molecule guests at room temperature, but how they self-assemble and what accelerates their assembly remain poorly understood. This article reports 81 distinct Octacid4•guest self-assembly pathways captured in unrestricted, unbiased molecular dynamics simulations. These pathways reveal that the self-assembly was initiated by the guest interaction with the cavity portal exterior of Octacid4 to increase the portal collisions that led to the portal expansion for guest ingress, and completed by the portal contraction caused by the guest docking inside the cavity to impede guest egress. The pathways also reveal that the self-assembly was accelerated by engaging populated host and guest conformations for the exterior interaction to increase the portal collision frequency. These revelations may help explain why the presence of an exterior binding site at the rim of the enzyme active site is a fundamental feature of fast enzymes such as acetylcholinesterase and why small molecules adopt local minimum conformations when binding to proteins. Further, these revelations suggest that irreversible noncovalent complexes with fast assembly rates could be developed—by engaging populated host and guest conformations for the exterior interactions—for materials technology, data storage and processing, molecular sensing and tagging, and drug therapy.

[1] Computer-Aided Molecular Design Laboratory, Mayo Clinic, Rochester, MN, USA. ✉email: camdl1@icloud.com

rreversible bimolecular complexes are highly desirable because (1) the residence of one molecule inside the cavity of another molecule is permanent until the breakdown of the cavity-containing molecule, (2) the complex formation is one-to-one stoichiometric, and (3) for drug therapy, the permanent residence and 1:1 stoichiometry confer the cavity-residing molecule with desired high metabolic stability, high potency-to-mass ratio, low-acting dose, and low off-target activity[1–4]. These complexes generally refer to irreversible covalent complexes that are self-assembled by two molecules between which there is a covalent bond resulting from the self-assembly process. The desirability of irreversible covalent complexes is apparent from the drug action mechanisms of aspirin[5], penicillin[6], and sotorasib[7], all of which involve an irreversible covalent complex. Notably, sotorasib was approved in May 2021 (https://www.fda.gov/news-events/press-announcements/fda-approves-first-targeted-therapy-lung-cancer-mutation-previously-considered-resistant-drug) as the first-in-class personalized treatment for a lung-cancer mutation previously considered resistant to drug therapy due to sotorasib's unique capability to clinically block the function of $KRAS^{G12C}$ (an enzyme mutant responsible for ~13% non-small-cell lung cancers) via irreversible complexation. However, the irreversible covalent complexes from in situ cysteine conjugation are limited by the infrequent presence of the noncatalytic cysteine in a protein cavity, and inhibiting $KRAS^{G12C}$ offers treatment for only a subset of cancer patients. A paradigm shift is needed for the design of irreversible bimolecular complexes.

In terms of both intrinsic binding from the thermodynamic perspective and constrictive binding from the kinetic perspective[8–10], the irreversible complexes include irreversible noncovalent complexes that are self-assembled by two molecules between which there is no covalent bond. Here, the intrinsic binding is the complexation governed by intermolecular interactions between the two molecules and between the solvent and each of the two molecules, while the constrictive binding is the complexation controlled by the thermal energy required for the guest to overcome the steric hindrance from the host during the assembly or disassembly process, as exemplified below by Cram's supramolecular capsules known as carcerand[11], hemicarcerand[12], and neocarcerand[13, 14].

The hallmark of the constrictive binding is the heightened energy barrier for the assembly or disassembly of a host•guest complex. This barrier can make the dissembled or assembled molecules kinetically stable, namely, it takes a long time to assemble the two disassembled molecules or disassemble the two assembled molecules if the barrier for the conversion is heightened. When the barrier is extremely high, assembly or disassembly requires covalent-bond making or breaking, respectively. For example, a guest can be noncovalently trapped in the cavity of the two bowl-shaped fragments that are rim-to-rim tethered by linkers (of a host known as carcerand) when the guest

is present in the reaction medium for the tethering reaction; the disassembly of the resulting carcerand complex is not allowed unless the host is broken by excessive heat[11]. When the barrier is moderately high, assembly or disassembly requires annealing. For example, a guest can enter or exit the cavity of a host known as hemicarcerand once the bimolecular system is heated to expand the cavity portal, and the guest remains inside the cavity once the system is cooled to contract the portal; the disassembly of the hemicarcerand complex is disallowed without heating[12, 15]. When the barrier is slightly high, assembly or disassembly can occur spontaneously and slowly at an ambient temperature. For some host•guest complexes, the disassembly can be disabled at the ambient temperature by the complexation that subsequently heightens the barrier for disassembly. For example, a guest can enter the cavity of a host known as neocarcerand at room temperature and remain in the cavity at room temperature unless the portal is opened by heating because the docking of the guest at the cavity induces a host conformational change that consequently closes all cavity portals[14].

The applications of carcerand and hemicarcerand are however limited because their complexes cannot be formed in situ and adiabatically. Although neocarcerand can form its complexes in situ and adiabatically[13, 14, 16], the applications of neocarcerand complexes are also limited due to their slow complexation rates. This underscores the need to understand how two molecules self-assemble into an irreversible noncovalent complex and what accelerates their assembly as these high-level questions hold the key to designing irreversible noncovalent bimolecular complexes with fast complexation rates for broad applications.

To promote irreversible noncovalent bimolecular complex design, this article reports 81 distinct pathways of the irreversible noncovalent self-assembly of neocarcerand Octacid4 with three known guests[13]—1,4-dioxane (dioxane), p-xylene (xylene), and naphthalene (Fig. 1 and Table S1). These pathways were captured in multiple distinct, independent, unrestricted, unbiased, and classical isobaric–isothermal molecular dynamics (MD) simulations at a high time resolution with an aggregated simulation time exceeding 3.761664 milliseconds at 298–363 K, rendering the structural and kinetic information needed to answer the two high-level questions and guide the design of irreversible noncovalent complexes that can be formed in situ and adiabatically with desired kinetics for materials technology, data storage and processing, molecular sensing and tagging, and drug therapy.

## Results

**The challenge of capturing self-assembly pathways.** In view of the current state of computational work on guest/ligand-binding pathways[17–20], unrestricted and unbiased MD simulations of the self-assembly of Octacid4 with its known guests are challenging to perform because the complexation times of Octacid4•guest were

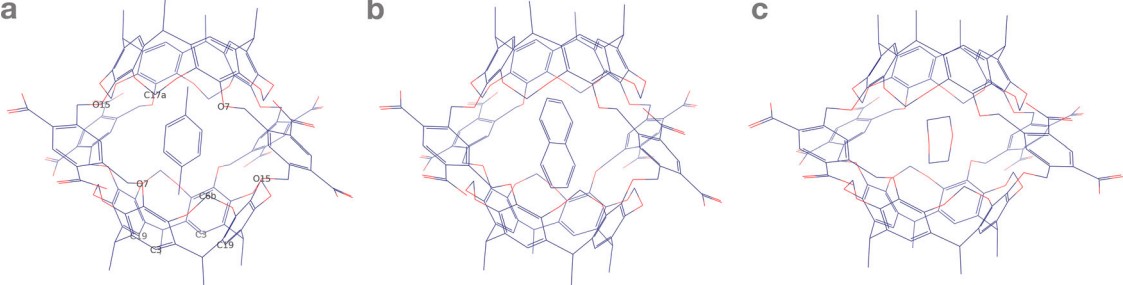

**Fig. 1 Structure of Octacid4 in complex with a small-molecule guest. a** Octacid4•p-xylene. **b** Octacid4•naphthalene. **c** Octacid4•1,4-dioxane. Carbon and oxygen are in blue and red, respectively. The axial or equatorial portal of Octacid4 comprises the C3 and C19 atoms or the O7, O15, C6b, and C17a atoms, respectively. Hydrogen and counter ion are not displayed for clarity.

estimated to be a few minutes (or hours for the bulkiest guest naphthalene) according to NMR experiments[13]. These complexation times are many orders of magnitude longer than current MD simulation times that are on the order of milliseconds.

To demonstrate the challenge of capturing the Octacid4•guest self-assembly pathways, multiple distinct, independent, unrestricted, unbiased, and classical isobaric–isothermal MD simulations (hereafter shortened to simulations) were performed using the fully deprotonated, *apo* Octacid4 that was surrounded by eight neutralizing sodium cations, 10 xylenes to mimic the use of the guest in 10 equivalent excess in the NMR experiments[13], 60 NaCl molecules to approximate the ionic strength of the experimental conditions[13], and 2359 TIP3P water molecules[21] to mimic the experimental aqueous solution[13]. Indeed, no autonomous complexation was observed in any set of 20 14,251.6-ns simulations at 298, 340, 363, or 370 K (Table S1) because the simulation time was many orders of magnitude shorter than the experimentally estimated complexation time for xylene.

**Using the phase-transfer catalyst to capture the self-assembly**. According to the reported NMR experiments[13], the experimentally observed self-assembly of Octacid4 with dioxane, xylene, or naphthalene in a sodium borate buffer at pH 9 is mechanistically driven by the sodium cation as a phase-transfer catalyst[22] that chelates with the carboxylates on the Octacid4 surface and consequently accumulates the guest on the host surface via the cation–π interaction[23] or the sodium chelation. The guest accumulation on the host surface is similar to immersing the sodium-chelated Octacid4 in a neat guest solution. Simulating the latter can substantially accelerate the self-assembly according to the law of mass action, enabling determination of relative complexation times of different guests with Octacid4 for mechanistic insights into the self-assembly. Simulating the latter also allows the use of linear-regression analysis to examine the convergency and internal consistency of the MD simulations. A goodness of fit ($r^2$) of <0.70 for the natural logarithm plot of the host population versus the simulation time indicates problematic simulations. This is because the first-order rate (viz., the exponential decay of the host population over the simulation time) is expected for simulating the self-assembly of Octacid4 with its guest in large excess.

Accordingly, 40 simulations were performed using the fully deprotonated, *apo* Octacid4 that was surrounded by eight neutralizing sodium cations and 150 xylenes. Here, the number of xylenes was arbitrarily chosen. Gratifyingly, all 40 14,251.6-ns simulations with the fully deprotonated, *apo* Octacid4 with 150 xylenes captured the self-assembly of Octacid4 with xylene at 298 K. Additional simulations were performed using each of the five variations: (1) replacing the fully deprotonated, *apo* Octacid4 with the octa-anionic Octacid4 possessing five water molecules inside the cavity, (2) increasing the number of xylenes to 250, (3) replacing 150 xylenes with 150 dioxanes, (4) replacing 150 xylenes with 150 naphthalenes, or (5) changing the Berendsen thermostat to the Langevin thermostat (Table S1). All these simulations captured the self-assembly event. For the simulations with the water-bound Octacid4, all water molecules inside the cavity had a high propensity to interact with the sodium cations and the carboxylates outside the cavity, and egression of all five water molecules occurred prior to the ingression of xylene. By contrast, no complexation was observed under the same conditions if the fully deprotonated, *apo* Octacid4 surrounded by 8 sodium cations was replaced with the fully protonated, *apo* Octacid4 without any cations (Table S1). All simulations described hereafter used the fully deprotonated, *apo*

Octacid4 surrounded with 8 sodium cations, 150 guests, and the Berendsen thermostat.

**Conformational characterization of the Octacid4•xylene self-assembly pathways**. Visual inspections of the 40 self-assembly pathways of Octacid4•xylene at 298 K revealed that all pathways comprised three common steps (Fig. 2 and Videos S1–S6 and S11–S16). In Step 1, xylene entered the cavity portal exterior space that was confined by two aromatic linkers of the host; as apparent from the representative videos, there were substantial collisions between xylene and the equatorial portal in the linker region of Octacid4 that led to the portal expansion for guest ingress. In Step 2, xylene passed one methyl group through the equatorial portal, then the phenyl group through the portal with the phenyl plane perpendicular to the axial axis (viz., the axis passing two axial portals in the bowl-shaped region), and last the other methyl group through the equatorial portal. In Step 3, xylene rotated ~90° to keep its phenyl plane parallel to the axial axis. This rotation caused the equatorial portal contraction and impeded guest egress, according to the reported structural analyses of *apo* Octacid4 and the Octacid4•xylene complex[14].

**Kinetic characterization of the Octacid4•xylene self-assembly pathways**. A survival analysis of the 40 14,251.6-ns simulations that all captured the self-assembly of Octacid4•xylene at 298 K showed xylene's mean complexation time to be 1022 ns (95% confidence interval: 750–1392 ns). Here the complexation time was defined as the first time instant at which xylene was inside the host cavity and had its phenyl plane parallel to the axial axis of the host. This xylene orientation was found in the most populated Octacid4•xylene conformation[14].

To dissect the self-assembly kinetics, the duration of the first step is herein termed priming time, and the duration of the last two steps is termed ingression time. These names are used because during the first step, the host and guest conformations are primed, through self-selection and conformational rearrangements, for guest ingress, and because during the last two steps, the guest enters the host cavity and rotates ~90° to form a complex.

The priming time was defined as a time period from the beginning of the MD simulation to the last time instant at which the distance between any guest hydrogen atoms and any host methylene hydrogen atoms was greater than 2.6 Å. The hydrogen–hydrogen distance cutoff (abbreviated as HH cutoff) was set at 2.6 Å for the following reasons. Visual inspection of all 40 Octacid4•xylene pathways revealed that xylene was outside of the Octacid4 cavity as long as the center-of-mass distance cutoff (abbreviated as COM cutoff) for Octacid4 and its guest was ≥7 Å. However, xylene could have its terminal methyl group contact the host cavity portal to slightly enter the host cavity portal in some of the 40 Octacid4•xylene pathways at the COM cutoff of 7 or 8 Å. Xylene could also be relatively away from the portal in some pathways at the COM cutoff of ≥8 Å. Therefore, rather than using the COM cutoff, the HH cutoff of 2.6 Å was used to avoid both the methyl group contacting the portal (which consequently shortens the ingression time) and the guest being away from the portal (which consequently lengthens the ingression time). The ingression time was defined as a time period from the last time instant at which the distance between any guest hydrogen atoms and any host methylene hydrogen atoms was greater than 2.6 Å to the first time instant at which the guest rotated ~90° inside the cavity. The complexation time is now a sum of the priming and ingression times.

This dissection reveals that the ingression time (8–934 ps; Table S2) is a fraction of the priming time (12–14,199 ns; Table S2), and hence the contribution of the ingression time to

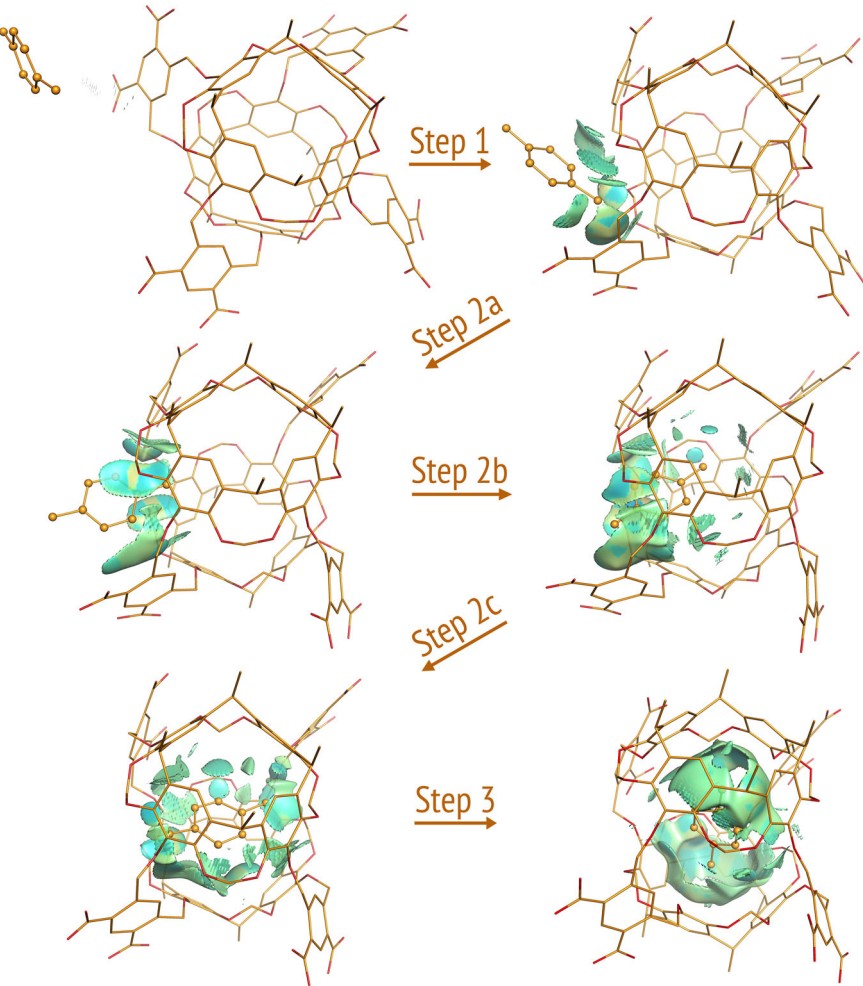

**Fig. 2 Three common steps of the 40 Octacid4•*p*-xylene self-assembly pathways at 298 K.** Octacid4 and *p*-xylene are in the stick and stick-and-ball models, respectively. Carbon and oxygen are in orange and red, respectively. Hydrogen, counter ion, and the *p*-xylenes in the bulk phase are not displayed for clarity. The noncovalent interaction gradient isosurfaces show the intermolecular interactions using a blue–red scale with blue indicating strong attractions and red indicating strong repulsions. All conformations shown here were obtained directly (no energy minimization) from Simulation 19 of the 40 14,251.6-ns simulations at 298 K. The isosurfaces show no repulsion but increasing attraction between the two molecules throughout the pathway.

the complexation time is inconsequential. In other words, the guest ingress is so fast that the complexation time is determined primarily by the priming time. This conclusion is independent of the use of the HH cutoff of 2.6 Å because changes of this cutoff only slightly affect the ingression time that is several orders of magnitude shorter than the priming time. For example, as apparent from Table S2, the priming and complexation times determined from the HH cutoff of 2.6 Å are identical to those determined from the COM cutoff of 10 Å, and the average ingression times from the HH cutoff versus the COM cutoff are 95 versus 105 ps.

**Relative complexation times of different guests with Octacid4.** Multiple simulations were performed for dioxane or naphthalene under the same simulation conditions as those for xylene. These simulations showed that the self-assembly of Octacid4 with dioxane or naphthalene was much slower than that of Octacid4•xylene according to the priming time defined using either the HH cutoff of 2.6 Å or the COM cutoff of 8 Å for dioxane and 10 Å for naphthalene (Table S2). The use of the HH cutoff was according to the Octacid4•guest-pathway analysis, which revealed that dioxane and naphthalene were close enough (without slightly entering the host cavity portal) to the portal at the HH cutoff of

2.6 Å. The use of two different COM cutoffs was because, unlike dioxane that was close enough to the portal at the COM cutoff of 8 Å, naphthalene could have its β-carbon atoms contact the portal at the COM cutoff of 8 Å or be relatively away from the portal in some pathways at the COM cutoff of ≥8 Å. Consistent with the complexation kinetics of xylene described above, the priming and complexation times of dioxane and naphthalene determined from the HH cutoff are also identical to those determined from the COM cutoff (Table S2), and the average ingression times determined from the HH cutoff versus COM cutoff are 29/18 versus 29/17 ps for dioxane at 298/340 K and 88/670/7790 versus 88/668/7790 ps for naphthalene at 298/340/363 K (Table S2).

For dioxane, 100 6320-ns simulations captured two self-assembly pathways at 298 K (priming and ingression times using the HH cutoff of 2.6 Å: 3668 and 5681 ns and 6 and 52 ps, respectively; Table S2), and 40 12,640-ns simulations at 340 K captured 24 self-assembly pathways (priming and ingression times using the HH cutoff of 2.6 Å: 34–2816 ns, 6–50 ps; Table S2). For naphthalene, 100 6320-ns simulations at 298 K captured one self-assembly pathway (priming and ingression times using the HH cutoff of 2.6 Å: 5721 ns and 88 ps, respectively; Table S2), 100 6320-ns simulations at 340 K captured four self-assembly pathways (priming and ingression times using the HH cutoff of 2.6 Å: 1865–4389 ns, and

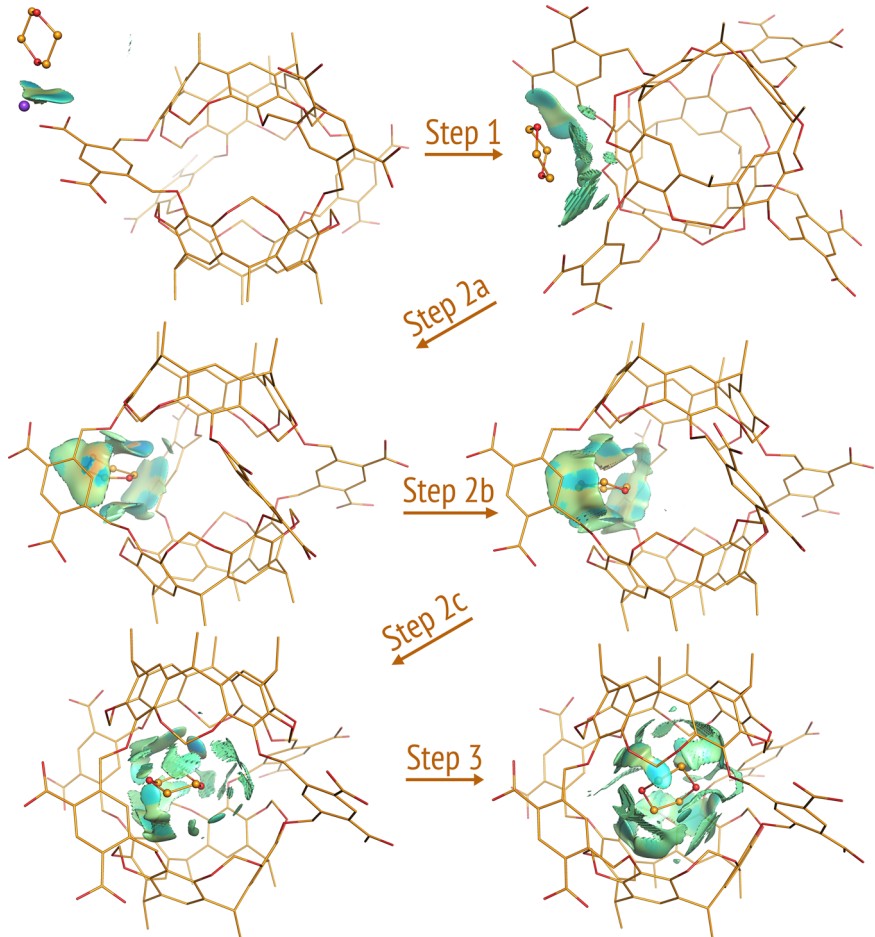

**Fig. 3 Three common steps of the 26 Octacid4•1,4-dioxane self-assembly pathways at 298 and 340 K.** Octacid4 and 1,4-dioxane are in the stick and stick-and-ball models, respectively. Carbon, oxygen, and sodium are in orange, red, and purple, respectively. Hydrogen, counter ion, and the 1,4-dioxanes in the bulk phase are not displayed for clarity, except for the ion that chelates 1,4-dioxane. The noncovalent interaction gradient isosurfaces show the intermolecular interactions using a blue–red scale with blue indicating strong attractions and red indicating strong repulsions. All conformations shown here were obtained directly (no energy minimization) from Simulation 76 of the 100 6320-ns simulations at 298 K. The isosurfaces reveal no repulsion but increasing attraction between the two molecules throughout the pathway.

16–2189 ps, respectively; Table S2), and 100 7900-ns simulations at 363 K captured ten self-assembly pathways (priming and ingression times using the HH cutoff of 2.6 Å: 651–7234 ns and 51–31,035 ps, respectively; Table S2). The ingression times of naphthalene were substantially longer than those of xylene and dioxane, but these longer ingression times of naphthalene were still a small portion of the naphthalene priming times, confirming that the ingression time is so short that the complexation time is governed largely by the priming time.

All pathways of dioxane and naphthalene shared the three common steps of the Octacid4•xylene pathways (Figs. 3 and 4), except for more profound collisions of dioxane or naphthalene with the equatorial portal than those of xylene in the first step (Videos S7–S10 and S17–S18) and subtle differences in the second step noted as follows. For dioxane, the second step involved the passing of the dioxane oxygen through the equatorial portal during which the oxygen was in the energetically less stable half-chair conformation, then the four-methylene portion of dioxane through the portal during which the methylene portion was in the energetically stable chair conformation, and last the other dioxane oxygen through the portal during which the oxygen was again in the half-chair conformation (Figs. 3 and 5 and Videos S7 and S17). Interestingly, the Octacid4•dioxane self-assembly process followed the mutually induced fitting

mechanism that is akin to the mechanism for a reported synthetic complex[24]. For naphthalene, the passing of the two head β-carbon atoms, then the α-carbon portion, and finally the tail β-carbon atoms of the guest through the host portal was completed at the second step (Fig. 4 and Videos S9–S10).

**Effect of conformational stability on complexation time.** To understand why the complexation time of xylene is much shorter than those of dioxane and naphthalene, conformational analyses of the 81 pathways were performed and revealed the involvement of three clusters of the Octacid4 conformations during the Octacid4•xylene self-assembly process at 298 K (Fig. 6). The most-populated cluster (population: 21/40) had two nearly orthogonal aromatic linkers that strongly interacted with xylene according to the intermolecular interactions depicted by the noncovalent interaction gradient isosurfaces[25] (Fig. 6a and Videos S1–S2 and S11–S12), and the assembly involving this cluster was captured mainly at the early stage of the simulations. The less-populated cluster (population: 12/40) had two nearly parallel, face-to-face aromatic linkers that moderately interacted with xylene (Fig. 6b and Videos S3–S4 and S13–S14), and the complexation involving this cluster was captured at the intermediate stage. The least-populated cluster (population: 7/40) had

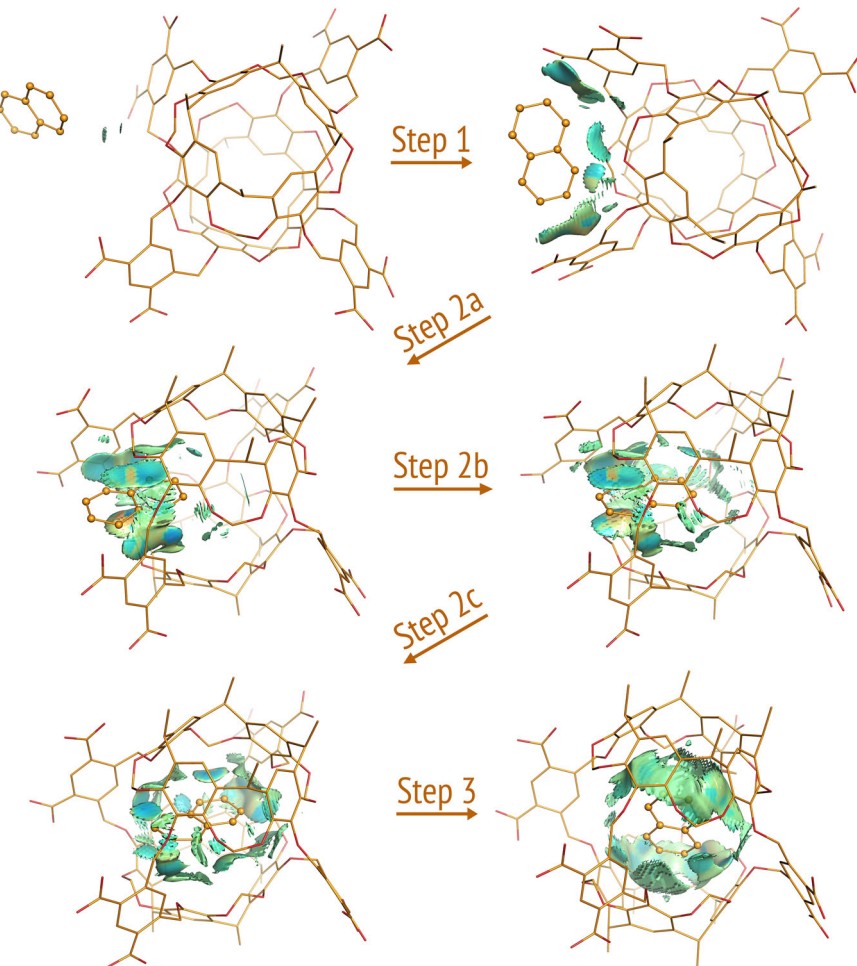

**Fig. 4 Three common steps of the 15 Octacid4•naphthalene self-assembly pathways at 298, 340, and 363 K.** Octacid4 and naphthalene are in the stick and stick-and-ball models, respectively. Carbon and oxygen are in orange and red, respectively. Hydrogen, counter ion, and the naphthalene in the bulk phase are not displayed for clarity. The noncovalent interaction gradient isosurfaces show the intermolecular interactions using a blue–red scale with blue indicating strong attractions and red indicating strong repulsions. All conformations shown here were obtained directly (no energy minimization) from Simulation 34 of the 100 6320-ns simulations at 298 K. The isosurfaces reveal no major repulsion but increasing attraction between the two molecules throughout the pathway.

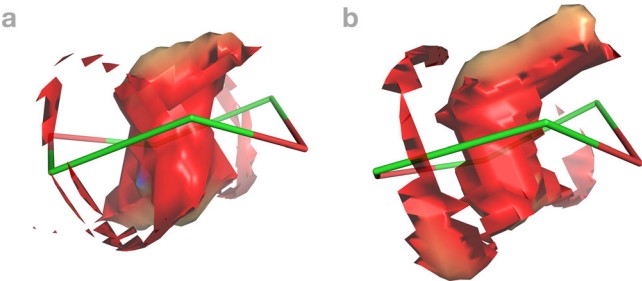

**Fig. 5 Noncovalent interaction gradient isosurfaces of 1,4-dioxane in different conformations. a** The chair conformation. **b** The half-chair conformation. Carbon and oxygen are in green and red, respectively. Hydrogen is not displayed for clarity. The gradient isosurfaces show the intramolecular interactions using a blue–red scale with blue indicating strong attractions and red indicating strong repulsions. All conformations shown here were obtained directly (no energy minimization) from Simulation 30 of the 100 6320-ns simulations at 298 K. The isosurfaces in panels **a** and **b** show stronger intramolecular repulsion in the half-chair conformation than that of the chair conformation.

two nearly-coplanar aromatic linkers that weakly interacted with xylene (Fig. 6c and Videos S5–S6 and S15–S16), and the complexation involving this cluster was captured at the late stage.

In contrast to xylene that had the attraction from the two nearly orthogonal aromatic linkers in its top-5 fastest pathways at 298 K (complexation times: 12–41 ns; Table S2; Figs. 2 and 6a), dioxane and naphthalene had the attraction from the two nearly parallel aromatic linkers in their fastest pathways at 298 K (complexation times: 3668 and 5729 ns; Table S2; Figs. 3 and 4). Consistent with the nature of π–π interactions[26], the Octacid4 conformation with two nearly orthogonal aromatic linkers corresponded to the 5th most populated conformation of *apo* Octacid4 in water, but the one with two nearly parallel, face-to-face aromatic linkers corresponded to none of the top-10 most populated conformations of the aqueous *apo* Octacid4.

These results demonstrate the effect of the host conformational stability on complexation time. More importantly, the results reveal that the self-assembly of Octacid4 with its guest is governed by the conformational complementarity between the two molecules not only during the ingression time but also during the priming time, and that the host or guest molecule can adopt, at a cost of lengthening the complexation time, an unpopulated

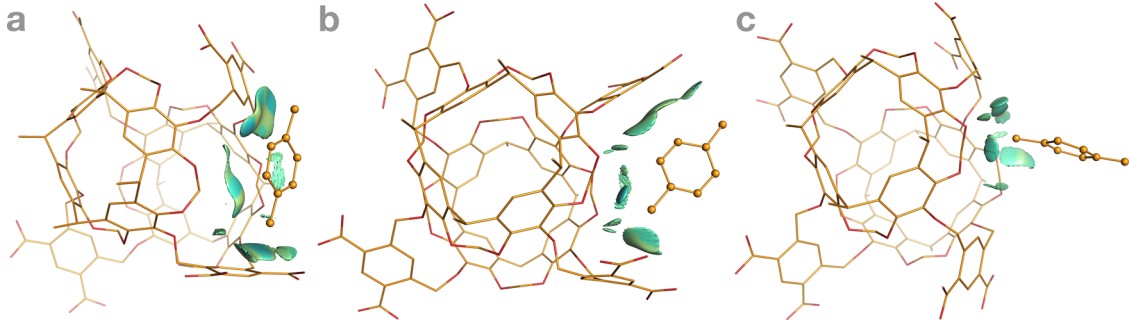

**Fig. 6 Three conformational clusters of the 40 Octacid4•*p*-xylene self-assembly pathways at 298 K. a** The most-populated cluster with two nearly orthogonal linkers that strongly attract *p*-xylene. **b** The less-populated cluster with two nearly parallel linkers that moderately attract *p*-xylene. **c** The least-populated cluster with two nearly-coplanar linkers that weakly attract *p*-xylene. The noncovalent interaction gradient isosurfaces show the intermolecular interactions using a blue–red scale with blue indicating strong attractions and red indicating strong repulsions. All conformations shown here were obtained directly (no energy minimization) from Simulations 17 for **a**, 7 for **b**, and 36 for **c** of the 40 14,251.6-ns simulations at 298 K.

conformation to proceed the self-assembly along a repulsion-free path when its populated conformation is not complementary to the conformation of its partner.

**Opposing effects of the sodium cation on complexation time.** According to the conformational analysis of the 43 self-assembly pathways of Octacid4 with dioxane (2 pathways), xylene (40 pathways), and naphthalene (1 pathway) at 298 K, the sodium cation that chelated with the Octacid4 carboxylate either coordinated with the guest oxygen atom or formed the cation–π interaction with the guest aromatic ring. However, the chelation with the oxygen atom of the guest that subsequently entered the host cavity was observed in one of the two pathways of dioxane (Table S2); the transient cation–π interaction with the aromatic ring of the cavity-entering guest was observed in only 20 of the 41 self-assembly pathways of xylene and naphthalene (Table S2). These observations indicate that the cavity-entering guest either does not interact at all or does not strongly interact with the carboxylate-chelated sodium cation, so that the cavity-entering guest is not trapped at the host linker region. Instead, the cavity-entering guest interacts with an immobilized guest, which is trapped in the linker region due to its strong interaction with the carboxylate-chelated sodium cation, via the π–π interaction for xylene or naphthalene or the van der Waals interaction for dioxane. Because the π–π interaction[26] and the van der Waals interaction are generally weaker than the cation–π interaction[23], these weak interactions enable the immobilized guest to usher the cavity-entering guest into the cavity, revealing the role of the sodium cation in shortening the complexation time of Octacid4 with dioxane, xylene, and naphthalene.

The conformational analysis of the 43 pathways also identified two small but interesting clusters of Octacid4 conformations that were derived a priori from the MD simulations (Fig. 7 and Videos S11–S18). In one cluster with a population of 11/43 (Table S2) that was associated mainly with the host conformations with two nearly parallel linkers, the carboxylates from two nearby linkers of Octacid4 formed a bidentate coordination with a sodium cation (Fig. 7a and Videos S11–S14 and S17–S18). This bidentate coordination rigidified a pair of the bidentate linkers and blocked one of the four equatorial portals of the host. In the other cluster with a population of 3/43 (Table S2) that was associated mainly with the host conformations with two nearly coplanar linkers, all four linkers of Octacid4 were involved in the bidentate coordination, resulting in rigidification of two pairs of the bidentate linkers and blockage of two of the four equatorial portals (Fig. 7b and Videos S15–S16). According to the energy

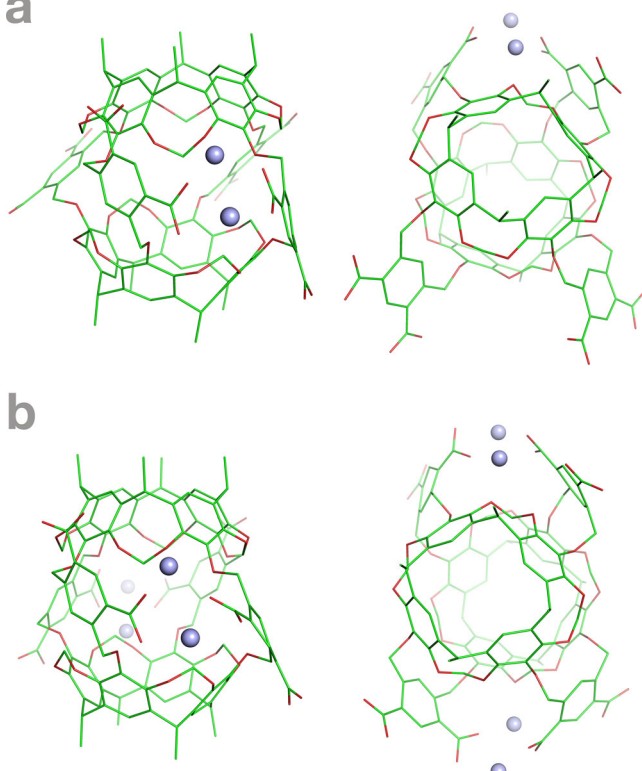

**Fig. 7 Sodium-restrained Octacid4 conformations. a** One pair of linkers that form bidentate coordination with the sodium cation. **b** Two pairs of linkers that form bidentate coordination with the sodium cation. Left: side view; Right: top view.

minimization and frequency calculations of the two simulation-derived conformations with one or two pairs of the bidentate linkers using the Gaussian 16 program and HF/6-31 G* or B3LYP/6-31 G*, the minimization-derived conformations closely resembled the simulation-derived conformations (heavy-atom root-mean-square deviations for one or two pairs of the bidentate linkers: 0.75 or 0.32 Å for HF/6-31 G* and 0.75 or 0.34 Å for B3LYP/6-31 G*), respectively, and no imaginary frequencies were found for the minimization-derived conformations, suggesting that the conformations with one or two pairs of the bidentate linkers derived a priori from the simulations were local minimal conformations. Because the host conformations

with the nearly parallel or nearly coplanar linkers were associated with long complexation times as described above, these results indicate that the sodium cation also played a role in lengthening the complexation time of Octacid4 with dioxane, xylene, and naphthalene and that the lengthening role is outweighed by the shortening role of the sodium cation as the populations of the host conformations with the bidendate linkers are lower than those of the host conformations without the bidendate linkers (Table S2).

## Discussion

**Internal and external consistencies of the self-assembly pathways.** The 81 self-assembly pathways described above exhibited internal or external consistencies as follows.

(1) The natural logarithm of the Octacid4 population versus the simulation time exhibited a linear relationship with $r^2$ of 0.98 for all 40 simulations that captured the Octacid4•xylene self-assembly, 0.85 for the first 20 of the 40 simulations, and 0.96 for the last 20 of the 40 simulations (Fig. 8). These $r^2$ values indicate the convergency and internal consistency of the 40 simulations for which the first-order self-assembly rate is expected as explained above.

(2) As described above, the simulations using the fully deprotonated Octacid4 under various conditions all captured the self-assembly of Octacid4 with dioxane, xylene, and naphthalene, but no autonomous complexation was observed under the same conditions if the fully deprotonated host was replaced by the fully protonated host. These results are consistent with the use of the Octacid4 solution containing the sodium borate buffer at pH 9 to detect the Octacid4•guest complexation in the NMR experiments[13] and with the phase-transfer catalysis undergirding those NMR experiments.

(3) One key finding of the present work is that the guest ingress is so fast that complexation time is determined primarily by the priming time. This finding is consistent with the report that many dense-phase reactions can be considered as gated reactions in that the rate of a local reaction is governed largely by the initial formation of a permissive atomic arrangement (viz., determined mainly by the system priming) within which the local transformation can proceed relatively rapidly[27].

(4) Conformational analyses of all 43 pathways of xylene, dioxane, and naphthalene at 298 K showed that Octacid4 adopted exclusively a cluster of V-shaped conformations to gulp its guests. These V-shaped conformations have the mean C17a–C6b distance (the distance between two diphenoxymethane carbon atoms that control the width of the cavity portal as shown in Fig. 1) of 7.0 Å (95% confidence interval: 7.0–7.1 Å) for the entrance portal and the corresponding mean C17a–C6b distance of 5.6 Å (95% confidence interval: 5.5–5.7 Å) for the opposing portal (Table S3). These mean distances are consistent with the reported V-shaped conformation proposed for the sliding-door mechanism for the gating of hemicarcerands[15] that are closely related to Octacid4.

(5) For the self-assembly at 298 K, 12 of the 40 captured pathways for xylene and all captured pathways for dioxane and naphthalene had two nearly parallel linkers that channeled the guest into the cavity (Table S2). These linker channels are consistent with the report that an antechamber formed by two parallel linkers of a hemicarcerand played a role in the gating of hemicarcerands[15].

(6) Relative to the noncovalent interaction gradient isosurfaces[25] of dioxane that showed the stronger intramolecular repulsion in the half-chair conformation than that in the chair conformation (Fig. 5), the isosurfaces of the Octacid4 complexes in the 43 self-assembly pathways of xylene (40), dioxane (2), and

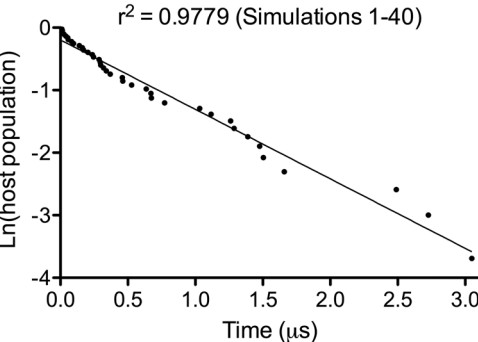

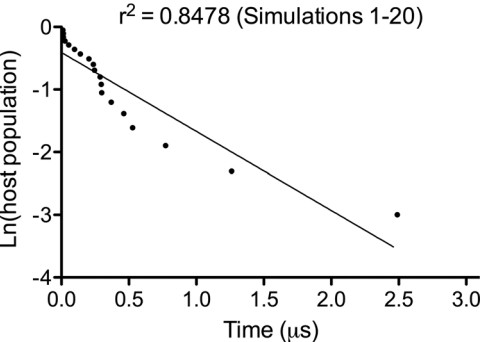

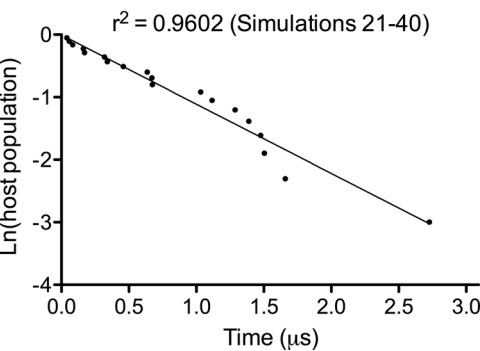

**Fig. 8 The exponential decay of the host population over the simulation time for the Octacid4•*p*-xylene self-assembly at 298 K.** The host population and simulation time were obtained from the 40 individual complexation times of *p*-xylene listed in Table S2. The linear-regression analysis was performed using the PRISM 5 program.

naphthalene (1) at 298 K revealed no repulsion (or only a trace of repulsion for naphthalene) but increasingly strong attraction between Octacid4 and its guest in the self-assembly process (Figs. 2–4). These observations are consistent with the report that the Corey–Pauling–Koltun space-filling models of xylene, dioxane, and naphthalene could be pushed (or pushed with effort for naphthalene) through portals of the space-filling model of Octacid4[13], indicating that a repulsion-free path exists for Octacid4 to self-assemble with xylene, dioxane, and naphthalene.

(7) Most importantly, the capturing of the self-assembly of Octacid4 with dioxane, xylene, and naphthalene in the simulations at 298 K is consistent with the Octacid4 complexation with these guests detected in the NMR experiments at 298 K[13].

**How Octacid4 self-assembles with its guests and what accelerates their assembly.** Collectively, the 81 pathways indicate that the irreversible noncovalent self-assembly process of Octacid4

with dioxane, xylene, and naphthalene was (1) initiated by the guest interaction with the cavity portal exterior of the host to increase the collisions of the portal that led to the portal expansion for guest ingress, (2) accelerated by engaging populated conformations of host and guest for the exterior intermolecular interaction to increase the portal collision frequency, and (3) completed by the portal contraction caused by the guest docking inside the cavity to impede guest egress. This type of self-assembly is a process of stepwise conformational rearrangements of the two molecules—kinetically starting from the pre-assembly state (at which the guest interacts with the exterior of the host) to the assembly and post-assembly states (at which the guest interacts partially and fully with the interior of the host, respectively)—to gradually increase the attraction between the two molecules along a repulsion–free path until the maximal attraction is reached. The kinetics of this process was governed primarily by the complementarity between the two molecules during the priming time (on the order of microseconds or longer) rather than the ingression time (on the order of picoseconds to nanoseconds).

**Heterogeneity of the Octacid4•xylene self-assembly pathways.** The 40 Octacid4•xylene self-assembly pathways derived from the converged MD simulations (as evident from Fig. 8) demonstrate the heterogeneity of the self-assembly pathways. As apparent from Table S2, at 298 K, xylene can enter the cavity of the three distinct clusters of the host conformations shown in Fig. 6 and Videos S1–S6. In each of these clusters, xylene can also enter the cavity of the host conformations with or without bidentate linkers (Videos S1–S6 and S11–S16), indicating that the Octacid4•xylene pathways can be shortened or lengthened by the sodium cation (Table S2).

This heterogeneity is akin to the heterogeneity of protein-folding pathways[28]. It points to a limitation of the simulation protocol used in this work because the protocol captured only the fast self-assembly pathways of Octacid4•dioxane and Octacid4•naphthalene at 298 K. This calls for the development of a new simulation protocol to avoid oversimplification of the self-assembly process such as equating the process to a few fast self-assembly pathways for Octacid4•dioxane and Octacid4•naphthalene. As apparent from the Octacid4•xylene self-assembly pathways, it is not a few complexation times (from the fast pathways) but the mean complexation time (from the fast, intermediate, and slow pathways) that offers insight into the delicate balance between the sodium cation's roles in lengthening and shortening the complexation time of Octacid4•xylene.

Given the known theoretical work on gated reactions[27] and the finding of the present work that the ingression time is so short that the complexation time is governed largely by the priming time, it is worth noting the need to avoid overcomplication of the self-assembly process, such as simulating the Octacid4•guest self-assembly using the octa-anionic Octacid4 possessing five water molecules inside the cavity or a more complicated host with a variable number of water molecules inside the cavity, for at least two reasons. First, it has not been determined experimentally whether the egression of water molecules that was observed in the present work will actually occur under the NMR experiment conditions once the water-bound Octacid4 is surrounded by a layer of water-insoluble xylene or naphthalene. Second, the water molecule is much smaller than dioxane, xylene, and naphthalene. The ingression and egression times of water are hence much shorter than those of the three guests, and binding of water molecules to the host cavity is likely opportunistic rather than intrinsic or constrictive. The inclusion of water molecules inside the cavity will affect the Octacid4•guest self-assembly but not

likely to the extent that the mean complexation time will be significantly altered.

**Design of irreversible noncovalent complexes with fast assembly rates.** Despite the heterogeneity issue as described above, the realization that the self-assembly initiation by the exterior intermolecular interaction and the control of the complexation time largely by the priming time is an advance and may help explain why the presence of a cryptic exterior site for substrate or inhibitor binding at the rim of the enzyme-active site is a fundamental feature of fast enzymes such as acetylcholinesterase[29–36] and why it is advantageous for the proapoptotic multidomain BAK protein to have a noncanonical BH3-binding groove[37] that abuts the canonical one for constrictive complexation of BAK—due to the steric hindrance from R88 and Y89 on the edge of the canonical groove[38]—with proapoptotic BH3-only proteins. The realization that the self-assembly acceleration by the adoption of populated conformations for the exterior intermolecular interaction is an advance and may help explain why small molecules reportedly prefer to adopt local minimum conformations when binding to proteins[39]. Further, these realizations suggest that irreversible noncovalent complexes with fast assembly rates could be designed by accounting for the complementarity between guest and host both of which adopt populated conformations for the exterior intermolecular interactions during the priming time. This design strategy may facilitate a paradigm shift from irreversible covalent complex design to irreversible noncovalent complex design for materials technology, data storage and processing, molecular sensing and tagging, and drug therapy—especially the personalized drug therapies for cancer patients with somatic mutations other than the $KRAS^{G12C}$ mutation[7].

## Methods

**Molecular dynamics simulation.** The fully deprotonated, *apo* Octacid4 neutralized with 8 sodium ions (or an Octacid4 in a different configuration, such as the fully protonated Octacid4, as listed in Table S1) was manually solvated with 150/250 copies of a guest (xylene, dioxane, or naphthalene) using PyMOL V1.7.0.3 (https://pymol.org) and tLEaP of the AmberTools 16 package (University of California, San Francisco) and then energy-minimized for 100 cycles of steepest-descent minimization followed by 900 cycles of conjugate-gradient minimization to remove close van der Waals contacts using SANDER of the AMBER 11 package (University of California, San Francisco), FF12MClm[40], and a cutoff of 8.0 Å for noncovalent interactions. The tLEaP input file for building the fully protonated Octacid4 and the Cartesian coordinates of the energy-minimized Octacid4•guest (in all configurations as listed in Table S1) are provided in Data S1 and S2, respectively. The energy-minimized system was slowly heated to 298/340/363/370 K in 30 steps under constant temperature and constant volume, and then equilibrated for $10^6$ timesteps under constant temperature of 298/340/363/370 K and constant pressure of 1 atm employing isotropic molecule-based scaling. Finally, a set of 20/40/100 distinct, independent, unrestricted, unbiased, and classical isobaric–isothermal MD simulations was performed for the resulting system using PMEMD of the AMBER 14/16/18/20 package (University of California, San Francisco), FF12MClm[40], and a periodic boundary condition at 1 atm and 298/340/363/370 K. All simulations used (i) a dielectric constant of 1.0, (ii) the Berendsen coupling algorithm[41] for thermostat and barostat, (iii) the particle mesh Ewald method[42] to calculate electrostatic interactions of two atoms at a separation of >8 Å, (iv) $\Delta t = 1.00$ fs of the standard-mass time[40, 43], (v) the SHAKE-bond-length constraint applied to all bonds involving hydrogen, (vi) a protocol to save the image closest to the middle of the "primary box" to the restart and trajectory files, (vii) a formatted restart file, (viii) the revised alkali-ion parameters[44], (ix) a cutoff of 8.0 Å for noncovalent interactions, (x) a uniform 10-fold reduction in the atomic masses of the entire simulation system (both solute and solvent)[40, 43], (xi) NTWX = 100 steps for coordinates' output, and (xii) default values of all other inputs of PMEMD.

Available in the Supporting Information of Ref. [43], FF12MClm is a revised AMBER protein forcefield with no parameterization for any Octacid4•guest complexes[40]. This forcefield is able to (1) capture the experimentally observed exponential decay of the non-native state population of fast-folding proteins over simulation time with $r^2 > 0.90$ and (2) fold these proteins with agreements between simulated and experimental folding times within factors of 0.6–1.4[45]. FF12MClm was used in this study to investigate the noncovalent self-assembly of small-molecule guests with Octacid4 whose aromatic linkers can flip between left- and right-handed configurations and usher the guest into the host cavity. This was

because of the effectiveness of FF12MClm in simulating the experimentally observed flipping between left- and right-handed configurations for C14–C38 of bovine pancreatic trypsin inhibitor in solution[40] and because of the need to compress the simulation time (viz., speed up simulations) by a factor of $10^{1/2}$ through 10-fold uniform reduction of the system mass. While the hydrogen mass repartitioning scheme can also speed up simulations, it was not used in this study because it would affect dynamic properties of the system[46]. The forcefield parameters for the fully deprotonated Octacid4, dioxane, xylene, and naphthalene are available in the Supplementary Information of Ref. [14]. The forcefield parameters for the neutral Octacid4 were developed using a published procedure[14] and provided in Data S1. The ab initio calculations for developing the fully protonated Octacid4 forcefield parameters were performed using Gaussian 98 (Revision A.7; Gaussian, Inc. Wallingford, CT). The chemical structures of HC2 and HCD2 and the assembly of HC2 into the fully protonated Octacid4 are shown in Fig. S1.

All MD simulations were performed using a dedicated cluster of 100 12-core Apple Mac Pros with Intel Westmere (2.40/2.93 GHz) and computers at the University of Minnesota Supercomputing Institute and the Mayo Clinic high-performance computing facility at the University of Illinois Urbana-Champaign National Center for Supercomputing Applications.

**Survival analysis**. The mean complexation time and its 95% confidence interval for the Octacid4•xylene self-assembly at 298 K was obtained from the 40 individual complexation times of xylene listed in Table S2 using the parametric survival function [the Surreg() function] implemented in the R survival package Version 3.2.0[47].

**Noncovalent interaction gradient isosurface**. All noncovalent interaction gradient isosurfaces were generated using the NCIPLOT program (Version 4)[48] with keyword RANGE (3, –0.1 to –0.02, –0.02 to 0.02, and 0.02–0.1 au) for Fig. 5 or keywords LIGAND (4.0 Å) and RANGE (3, –0.1 to –0.02, –0.02 to 0.02, and 0.02–0.1 au) for Figs. 2–4 and 6.

**Gaussian energy minimization and frequency calculations**. All energy minimization and frequency calculations were performed using Gaussian 16 (Revision C.01; Gaussian, Inc. Wallingford, CT) and HF/6-31 G* or B3LYP/6-31 G* on computers from the Mayo Clinic high-performance computing facility at the University of Illinois Urbana-Champaign National Center for Supercomputing Applications. Each frequency calculation was done using %mem = 40 Gb and %nproc = 16. The Cartesian coordinates of the Octacid4 conformations (with one or two pairs of bidentate linkers) before and after the minimization are provided in Data S3.

**Video**. The time series of the Cartesian coordinates of all heavy atoms of Octacid4 and the guest (that was initially outside of the Octacid4 cavity and then entered and remained inside the cavity) was extracted from the output file of an MD simulation and saved to a concatenated PDB file using the CPPTRAJ of the AMBER 14/16/18 package. The guest involved in the complexation was identified through the analysis of eight intermolecular distances between a guest atom (C3 for xylene, O1 for dioxane, and C1 for naphthalene) and the eight diphenoxymethane carbon atoms of Octacid4. The eight distances for each of 150 guests were calculated from the output file using the CPPTRAJ, and the guest was considered (confirmed by visual inspection) to be inside the Octacid4 cavity if all eight distances were <9.0 Å. The concatenated PDB file was then converted to a set of PNG files using the PyMOL V1.7.0.3 showing the top or side view of the bimolecular complex with the guest in the stick-and-ball model and the host in the stick model. Before the PDB-to-PNG conversion, each complex conformation was superimposed over the proceeding one using the PyMOL. The resulting PNG files was lastly converted to a video file using Adobe Photoshop 2021 (https://www.adobe.com).

## Data availability
Fig. S1, Tables S1–S3, Videos S1–S18, and Data S1–S3 are provided in the Supplementary Information. Other relevant data are available from the corresponding author upon reasonable request.

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

## Acknowledgements

This work was supported by the US Army Research Office (W911NF-16-1-0264) and the Mayo Foundation for Medical Education and Research. Responsibility for the information and views in this study lies entirely with the author. The author acknowledges the computing resources provided by the University of Minnesota Supercomputing Institute and the Mayo Clinic high-performance computing facility at the University of Illinois Urbana-Champaign National Center for Supercomputing Applications.

## Author contributions

Conceptualization, methodology, investigation, visualization, funding acquisition, project administration, supervision, and writing: Y.P.P.

## Competing interests

The author declares no competing interests.
