## [Peer Review File · Communications Chemistry]

Reviewers' comments:

Reviewer #1 (Remarks to the Author):

The paper addresses the important problem of noncovalent self-assembly of molecules with MD simulations of Octacid4 and guests. The MD simulations are well designed and reveal possible mechanism of the different stages of the assembly processes. Notably the simulations not only confirm the importance of the sodium cation in the assembly processes but also provide valuable insights on the specific structural and configurational details of the assembly processes in the presence of the sodium cation, information that are difficult to obtain experimentally.

Given the key roles that sodium cations play in the assembly process, it is highly desirable if the author could provide more details on the specific roles of the sodium cations, other than that the ions' presence are necessary for the self-assembly. Some basic but fundamental information include, for example, what conformational changes the ions induce that made the assembly possible or is it a=mostly effects of energetics changes with the presence of the sodium ions.

Reviewer #2 (Remarks to the Author):

In this work, the constrictive binding of neocarcerand Octacid4 with three guests—1,4-dioxane (dioxane), p-xylene (xylene), and naphthalene was studied by MD simulations at the MM level based on the experimental work of Yoon and Cram et al. (Ref. 13 in the paper). 40, 26 and 6 distinct Octacid4•guest self-assembly pathways was captured for Octacid4 with dioxane, xylene, and naphthalene, respectively. For the latter two guests, higher temperatures are needed to promote the complexation. These pathways comprised three common steps, among which the priming step is the longest and determines the total complexation time. These simulations showed that the self-assembly of Octacid4 with dioxane or naphthalene was much slower than that with xylene mainly due to the different stabilities of the conformations involved in the host-guest assembly. After the critical review of the draft, my major concerns are:

(1) The title seems too general considering the limited numbers of the systems studied. In addition, the work is mainly on the three special host-guest systems, however many discussions are concerned with protein-ligand systems.

(2) As mentioned in the paper, the sodium cation that chelates with the carboxylates on the Octacid4 surface is essential for the complexation of Octacid4 with three guests. It seems that this chelation may have some influence on the host conformation, however this is not explored in the work.

(3) The whole self-assembling process was divided into three steps: priming, ingress and complexation. It seems that the priming step is not strictly defined, in particular, when does this step initiate?

A few minor issues: the title of Figure 1 contains some typos; on page 4, "8-934 ps" should be 8-934 ns.

Overall speaking, the paper is well written and the data is substantial and well organized and should be suitable for publication at Commun. Chem. after some further improvement.

Reviewer #3 (Remarks to the Author):

In the present manuscript, the author used molecular dynamics (MD) simulations to study the non-covalent irreversible self-assembly process of a series of guests into a supramolecular capsule. MD simulations of the neocarcerand octacid immersed in a guest solution provided a number of guest binding pathways inside the capsule cavity that unravel the molecular basis of the self-assembly process. This work represents a continuation of a previous study (Communications Chemistry volume 4, Article number: 26 (2021)) where the intrinsic dynamics of the formed host-guest complex was studied in water solution. Briefly, from my point of view the results presented are interesting and provide relevant information for understanding the self-assembly process of these supramolecular complexes. My major concerns are that the generalizations to enzyme-inhibitor complexes are a bit too ambitious and that the simulation conditions used in the MD simulations cannot be directly comparable with experimental results. Moreover, there are some studies of guest-binding pathways in supramolecular hosts that should be referenced in the present study. I have some other comments/suggestions that the authors should clarify.

Major concerns:

1. From my point of view, the manuscript requires reorganizing some parts and the inclusion of additional references in the discussion.

1a. First, the article should be divided into different sections to facilitate the comprehension (introduction, methods, results, define different sections inside results, ...). Sometimes it is not clear when one section starts and finishes the other. It would help to separate the results section into different sub-sections.

1b. Second, in the introduction and in the final discussion a lot of emphasis is put in generalizing the findings for this particular host-guest supramolecular complex to enzyme-inhibitor complexes. From my point of view, the findings of the present manuscript are interesting in terms of supramolecular host-guest binding but generalizing these observations to the formation of protein-ligand complexes is a bit too ambitious. In this context, there are a lot of works and methodologies to study the thermodynamics and kinetics of protein-ligand binding pathways (see for example WIREs Comput Mol Sci. 2020; 10:e1455 for a summary of different methodologies). With these methodologies it has been possible to unravel the molecular basis of ligand binding for a variety of systems describing how distinct protein conformation and cryptic pockets arise along the binding pathway. The author should consider some of these works when generalizing the findings to enzyme-inhibitor complexes.

1c. Finally, there are some studies of guest binding and unbinding in the supramolecular complexes (see for example SAMPL6 challenge (where an octa acid Gibb cavitand was used (Journal of Computer-Aided Molecular Design volume 32, 1001–1012 (2018))) or J. Chem. Theory Comput. 2019, 15, 4, 2433–2443 and J. Chem. Theory Comput. 2020, 16, 9, 5526–5547 using cyclodextrins as a reference). Molecular dynamics simulations and enhanced sampling techniques have been extensively used to characterize ligand-binding into supramolecular complexes. The author should contextualize the findings of the present manuscript with respect to the previous works in this area.

2. The role of sodium cations in the self-assembly process is not discussed. In the introduction and in the beginning of the results sections it is mentioned that sodium cations accelerate the binding process. However, sodiums are not included in the Figures and their potential cation- π interaction with the guest is not discussed. The authors should include a brief discussion on the role of sodium

cations in the MD simulations and also analyze the cation- π interactions in the NCI calculations.

3. The MD simulations are performed with a guest solution and an empty cavity. How do these conditions compare with experiments? Using these conditions clearly accelerates the binding process, however, it can also alter the kinetics and the molecular basis of the self-assembly process in comparison to experimental conditions (e.g. induced-fit vs conformational selection pathways). Performing the MD simulations with different guest solutions can alter the conformational ensemble of the host in comparison with the conformations accessible in water. The author should comment in more detail how these two conditions can impact the outcome of the guest binding pathways and kinetics obtained from the MD simulations. In the manuscript it is mentioned that a system that includes water molecules inside the host cavity is simulated as a test. However, these results are not described in the main text. The presence of solvent molecules inside the cavity can alter the kinetics of guest binding.

Minor concerns:

4. From my point of view the title of the manuscript is too general. The title should be more focused on the supramolecular system studied.

5. A table with the complete list of systems simulated with MD Simulations should be included in the supporting information. A lot of tests are performed with a different number of guests, conditions, ... However, these tests are not mentioned in the methods section or in the supporting information. It would be good to summarize all these tests in a table in the SI.

6. A general scheme of the systems simulated with the supramolecular capsule and the different guests should be included in the main text.

7. In the abstract and in the main text, the author uses the terminology millisecond MD simulations. However, independent microsecond MD simulations are performed. Considering that the MD trajectories are analyzed independently and do not exceed the microsecond time scale I would suggest removing the word millisecond, otherwise it can be misleading.

8. Additionally, to complement the figures showing the encapsulation process, the author can consider adding plots that monitor the distance between the center of mass of the capsule and the center of mass of the guest with the different steps described in figure 2 highlighted in the plot. In this way, the reader can put together the molecular basis of the self-assembly process with the simulation time.

Response to the reviewers' comments

I greatly appreciate the three reviewers' precious time and their insightful comments that have helped improve the scientific rigidity and clarity of the manuscript. Below is my point-to-point response to their comments. The changes made according to their comments are colored in red in the revised manuscript.

Reviewer 1 (Remarks to the Author):

The paper addresses the important problem of noncovalent self-assembly of molecules with MD simulations of Octacid₄ and guests. The MD simulations are well designed and reveal possible mechanism of the different stages of the assembly processes. Notably the simulations not only confirm the importance of the sodium cation in the assembly processes but also provide valuable insights on the specific structural and configurational details of the assembly processes in the presence of the sodium cation, information that are difficult to obtain experimentally.

Given the key roles that sodium cations play in the assembly process, it is highly desirable if the author could provide more details on the specific roles of the sodium cations, other than that the ions' presence are necessary for the self-assembly. Some basic but fundamental information include, for example, what conformational changes the ions induce that made the assembly possible or is it a mostly effects of energetics changes with the presence of the sodium ions.

Response: I have performed new calculations and added two paragraphs at the end of the Results section describing the two opposing effects of the sodium cation on complexation time. I found that the sodium cation could function as a phase-transfer catalyst to build up the local concentration of the guest at the cavity portal, which in turn shortens the complexation time. I also found that the sodium ion could rigidify the host conformations through bidentate coordination with two nearby linkers and in turn lengthen the complexation time. However, the lengthening role was outweighed by the shortening role of the sodium cation. These observations were based on a conformational analysis of the 43 self-assembly pathways at 298 K and energy minimization and frequency calculations of the sodium-chelating Octacid₄ conformations (derived a priori from the MD simulations) using the Gaussian 16 program and the HF/6-31G* or B3LYP/6-31G* basis set. I have accordingly expanded the Methods section, increased the number of video files from 10 to 18, provided a figure to show two host conformations rigidified by the sodium cation (Fig. 7), and substantially expanded Table S1 to include the information on bidentate coordination, the sodium cation-pi interaction with xylene/naphthalene, and the sodium chelation with dioxane.

Reviewer 2 (Remarks to the Author):

*In this work, the constrictive binding of neocarcerand Octacid₄ with three guests—1,4-dioxane (dioxane), *p*-xylene (xylene), and naphthalene was studied by MD simulations at the MM level based on the experimental work of Yoon and Cram et al. (Ref. 13 in the paper). 40, 26 and 6 distinct Octacid₄•guest self-assembly pathways was captured for Octacid₄ with dioxane, xylene, and naphthalene, respectively. For the latter two guests, higher temperatures are needed to promote the complexation. These pathways comprised three common steps, among which the priming step is the*

longest and determines the total complexation time. These simulations showed that the self-assembly of Octacid₄ with dioxane or naphthalene was much slower than that with xylene mainly due to the different stabilities of the conformations involved in the host-guest assembly. Overall speaking, the paper is well written and the data is substantial and well organized and should be suitable for publication at Commun. Chem. after some further improvement.

Major concerns

(1) The title seems too general considering the limited numbers of the systems studied. In addition, the work is mainly on the three special host-guest systems, however many discussions are concerned with protein-ligand systems.

Response: I have changed the title to “How two molecules self-assemble into a noncovalent irreversible complex and what accelerates the assembly” to reflect the scope of the present work on noncovalent irreversible complexation. I have also toned down the generalization to protein-ligand complexes with the following changes.

Previous

“... the realization that the self-assembly initiation by the exterior intermolecular interaction and the control of the complexation time largely by the priming time **explains** why the presence of a cryptic exterior site for substrate or inhibitor binding at the rim of the enzyme active site is a fundamental feature of fast enzymes such as acetylcholinesterase²⁵⁻³² and why it is advantageous for the proapoptotic multidomain BAK protein to have a noncanonical BH₃-binding groove³³ that abuts the canonical one for constrictive complexation of BAK—due to the steric hindrance from R88 and Y89 on the edge of the canonical groove³⁴—with proapoptotic BH₃-only proteins. The realization that the self-assembly acceleration by the adoption of populated conformations for the exterior intermolecular interaction **explains** why small molecules reportedly prefer to adopt local minimum conformations when binding to proteins³⁵. ...”

Current

“... the realization that the self-assembly initiation by the exterior intermolecular interaction and the control of the complexation time largely by the priming time **is an advance and may help explain** why the presence of a cryptic exterior site for substrate or inhibitor binding at the rim of the enzyme active site is a fundamental feature of fast enzymes such as acetylcholinesterase²⁵⁻³² and why it is advantageous for the proapoptotic multidomain BAK protein to have a noncanonical BH₃-binding groove³³ that abuts the canonical one for constrictive complexation of BAK—due to the steric hindrance from R88 and Y89 on the edge of the canonical groove³⁴—with proapoptotic BH₃-only proteins. The realization that the self-assembly acceleration by the adoption of populated conformations for the exterior intermolecular interaction **is an advance and may help explain** why small molecules reportedly prefer to adopt local minimum conformations when binding to proteins³⁵. ...”

(2) As mentioned in the paper, the sodium cation that chelates with the carboxylates on the Octacid₄ surface is essential for the complexation of Octacid₄ with three guests. It seems that this chelation may have some influence on the host conformation, however this is not explored in the work.

Response: Indeed, the sodium chelation induced two small clusters of relatively rigid host conformations that were identified a priori from the MD simulations. Subsequent energy minimization and frequency calculations using the Gaussian 16 program and the HF/6-31G* or B₃LYP/6-31G* basis set suggested that these rigid conformations from the simulations were local minimal conformations and played a role in lengthening complexation time. However, this lengthening role is outweighed by the role of the sodium cation as a phase-transfer catalyst in shortening the complexation time. I have accordingly provided two paragraphs at the end of the Results section describing the two opposing effects of the sodium cation on complexation time, expanded the Methods section, increased the number of the video files from 10 to 18, provided a figure to show two host conformations rigidified by the sodium cation (Fig. 7), and substantially expanded Table S1 to include the information on bidendate coordination, the sodium cation- π interaction with xylene or naphthalene, and the sodium chelation with dioxane.

(3) The whole self-assembling process was divided into three steps: priming, ingress and complexation. It seems that the priming step is not strictly defined, in particular, when does this step initiate?

Response: I have added the following definition of the priming and ingress times in the paragraph describing the kinetic characterization.

“... The priming time was defined as a time period from the beginning of the MD simulation to the last time instant at which the distance between any guest hydrogen atoms and any host methylene hydrogen atoms was greater than 2.6 Å. The ingress time was defined as a time period from the last time instant at which the distance between any guest hydrogen atoms and any host methylene hydrogen atoms was greater than 2.6 Å to the first time instant at which the guest rotated $\sim 90^\circ$ inside the cavity. ...”

Minor issue

The title of Figure 1 contains some typo.

Response: I have corrected the typographical error.

On page 4, “8-934 ps” should be 8-934 ns.

Response: I have double checked the raw data and confirm that the ingress time is indeed 8-934 ps, which is many orders of magnitude smaller than the priming time.

Reviewer 3 (Remarks to the Author):

In the present manuscript, the author used molecular dynamics (MD) simulations to study the non-covalent irreversible self-assembly process of a series of guests into a supramolecular capsule. MD simulations of the neocarcarand octacid immersed in a guest solution provided a number of guest binding pathways inside the capsule cavity that unravel the molecular basis of the self-assembly process. This work represents a continuation of a previous study (Communications Chemistry volume 4, Article number: 26 (2021)) where the intrinsic dynamics of the formed host-guest complex

was studied in water solution. Briefly, from my point of view the results presented are interesting and provide relevant information for understanding the self-assembly process of these supramolecular complexes. My major concerns are that the generalizations to enzyme-inhibitor complexes are a bit too ambitious and that the simulation conditions used in the MD simulations cannot be directly comparable with experimental results. Moreover, there are some studies of guest-binding pathways in supramolecular hosts that should be referenced in the present study. I have some other comments/suggestions that the authors should clarify.

Response: See my responses to the following eight comments that address the ones noted above.

Major concerns

1. From my point of view, the manuscript requires reorganizing some parts and the inclusion of additional references in the discussion.

1a. First, the article should be divided into different sections to facilitate the comprehension (introduction, methods, results, define different sections inside results, ...). Sometimes it is not clear when one section starts and finishes the other. It would help to separate the results section into different sub-sections.

Response: I have changed the manuscript format accordingly.

1b. Second, in the introduction and in the final discussion a lot of emphasis is put in generalizing the findings for this particular host-guest supramolecular complex to enzyme-inhibitor complexes. From my point of view, the findings of the present manuscript are interesting in terms of supramolecular host-guest binding but generalizing these observations to the formation of protein-ligand complexes is a bit too ambitious. In this context, there are a lot of works and methodologies to study the thermodynamics and kinetics of protein-ligand binding pathways (see for example WIREs Comput Mol Sci. 2020; 10:e1455 for a summary of different methodologies). With these methodologies it has been possible to unravel the molecular basis of ligand binding for a variety of systems describing how distinct protein conformation and cryptic pockets arise along the binding pathway. The author should consider some of these works when generalizing the findings to enzyme-inhibitor complexes.

Response: I have made the following changes to avoid the generalization to enzyme-inhibitor complexes. As noted in my next response I have also cited the WIREs Comput Mol Sci paper.

Previous

“Abstract: ... These revelations explain why the presence of an exterior binding site at the rim of the enzyme active site is a fundamental feature of fast enzymes such as acetylcholinesterase and why small molecules adopt local minimum conformations when binding to proteins. ...”

“Discussion: ... the realization that the self-assembly initiation by the exterior intermolecular interaction and the control of the complexation time largely by the priming time **explains** why the presence of a cryptic exterior site for substrate or inhibitor binding at the rim of the enzyme active site is a fundamental feature of fast enzymes such as acetylcholinesterase²⁵⁻³² and why it is advantageous for the proapoptotic multidomain BAK protein to have a noncanonical BH₃-binding groove³³ that abuts the canonical one for constrictive complexation of BAK—due to the steric hindrance from R88 and Y89 on the edge of the canonical groove³⁴—with proapoptotic

BH₃-only proteins. The realization that the self-assembly acceleration by the adoption of populated conformations for the exterior intermolecular interaction **explains** why small molecules reportedly prefer to adopt local minimum conformations when binding to proteins³⁵. ...”

Current

“Abstract: ... These revelations **may help explain** why the presence of an exterior binding site at the rim of the enzyme active site is a fundamental feature of fast enzymes such as acetylcholinesterase and why small molecules adopt local minimum conformations when binding to proteins. ...”

“Discussion: ... the realization that the self-assembly initiation by the exterior intermolecular interaction and the control of the complexation time largely by the priming time **is an advance and may help explain** why the presence of a cryptic exterior site for substrate or inhibitor binding at the rim of the enzyme active site is a fundamental feature of fast enzymes such as acetylcholinesterase²⁵⁻³² and why it is advantageous for the proapoptotic multidomain BAK protein to have a noncanonical BH₃-binding groove³³ that abuts the canonical one for constrictive complexation of BAK—due to the steric hindrance from R88 and Y89 on the edge of the canonical groove³⁴—with proapoptotic BH₃-only proteins. The realization that the self-assembly acceleration by the adoption of populated conformations for the exterior intermolecular interaction **is an advance and may help explain** why small molecules reportedly prefer to adopt local minimum conformations when binding to proteins³⁵. ...”

1c. Finally, there are some studies of guest binding and unbinding in the supramolecular complexes (see for example SAMPL6 challenge (where an octa acid Gibb cavitand was used (Journal of Computer-Aided Molecular Design volume 32, 1001–1012 (2018))) or J. Chem. Theory Comput. 2019, 15, 4, 2433–2443 and J. Chem. Theory Comput. 2020, 16, 9, 5526–5547 using cyclodextrins as a reference). Molecular dynamics simulations and enhanced sampling techniques have been extensively used to characterize ligand-binding into supramolecular complexes. The author should contextualize the findings of the present manuscript with respect to the previous works in this area.

Response: I have cited the above articles including the WIREs Comput Mol Sci review article as Refs. 17–20. I have also added a subsection in Discussion on the heterogeneity of the Octacid₄•xylene self-assembly pathways to discuss the need and challenge to capture the fast, slow, and intermediate reaction pathways from the perspective of the sodium’s dual effects on complexation time.

2. The role of sodium cations in the self-assembly process is not discussed. In the introduction and in the beginning of the results sections it is mentioned that sodium cations accelerate the binding process. However, sodiums are not included in the Figures and their potential cation- π interaction with the guest is not discussed. The authors should include a brief discussion on the role of sodium cations in the MD simulations and also analyze the cation- π interactions in the NCI calculations.

Response: I have performed new calculations and added two paragraphs at the end of the Results section describing the two opposing effects of the sodium cation on complexation time. Briefly, the sodium cation can rigidify the Octacid₄ conformation and play a role in lengthening

complexation time, in addition to the role in shortening the complexation time, but the lengthening role is outweighed by the shortening role. I have accordingly expanded the Methods section, increased the number of the videos from 10 to 18, provided a figure (Fig. 7) to show two host conformations rigidified by the sodium cation, and substantially expanded Table S1 to include the information on bidentate coordination, the sodium cation- π interaction with xylene/naphthalene, and the sodium chelation with dioxane.

3. The MD simulations are performed with a guest solution and an empty cavity. How do these conditions compare with experiments? Using these conditions clearly accelerates the binding process, however, it can also alter the kinetics and the molecular basis of the self-assembly process in comparison to experimental conditions (e.g. induced-fit vs conformational selection pathways). Performing the MD simulations with different guest solutions can alter the conformational ensemble of the host in comparison with the conformations accessible in water. The author should comment in more detail how these two conditions can impact the outcome of the guest binding pathways and kinetics obtained from the MD simulations. In the manuscript it is mentioned that a system that includes water molecules inside the host cavity is simulated as a test. However, these results are not described in the main text. The presence of solvent molecules inside the cavity can alter the kinetics of guest binding.

Response: I have accordingly provided additional information for the simulations of the water-bound Octacid₄ in the second subsection of the Results section (see below).

“... For the simulations with the water-bound Octacid₄, all water molecules inside the cavity had a propensity to coordinate with the sodium cations outside the cavity, and egression of all five water molecules occurred prior to the ingress of xylene. ...”

I have also added a subsection on the heterogeneity of the Octacid₄•xylene self-assembly pathways in Discussion. This subsection notes the avoidance of overcomplication of the self-assembly process such as simulating the Octacid₄•guest self-assembly using the octa-anionic Octacid₄ possessing 5 water molecules inside the cavity or a more complicated host with a variable number of water molecules inside the cavity for at least two reasons. First, there is no experimental evidence to show the egression of water molecules observed in the present work will not occur under the NMR experimental conditions once the water-bound Octacid₄ is surrounded by a layer of water-insoluble xylene or naphthalene. Second, the water molecule is small and hence its binding to the host cavity is likely opportunistic. Therefore, inclusion of water molecules inside the Octacid₄ cavity will affect the self-assembly of Octacid₄•guest but not likely to significantly change mean complexation time.

Minor concerns

4. From my point of view the title of the manuscript is too general. The title should be more focused on the supramolecular system studied.

Response: I have changed the title to “How two molecules self-assemble into a noncovalent irreversible complex and what accelerates the assembly” to reflect the scope of the present work on noncovalent irreversible complexation of neocarcarand Octacid₄.

5. A table with the complete list of systems simulated with MD Simulations should be included in the supporting information. A lot of tests are performed with a different number of guests, conditions, ... However, these tests are not mentioned in the methods section or in the supporting information. It would be good to summarize all these tests in a table in the SI.

Response: I have provided a complete list of the simulated systems and their conditions and results in Table S1 of the revision.

6. A general scheme of the systems simulated with the supramolecular capsule and the different guests should be included in the main text.

Response: I have described a general scheme of the simulated systems in the main text in the last paragraph of Introduction that reads as follows:

“To promote noncovalent irreversible bimolecular complex design, this article reports 81 distinct pathways of the noncovalent irreversible self-assembly of neocarcerand Octacid₄ with three known guests¹³—1,4-dioxane (dioxane), *p*-xylene (xylene), and naphthalene (Fig. 1 and Table S1).”

7. In the abstract and in the main text, the author uses the terminology millisecond MD simulations. However, independent microsecond MD simulations are performed. Considering that the MD trajectories are analyzed independently and do not exceed the microsecond time scale I would suggest removing the word millisecond, otherwise it can be misleading.

Response: I have removed “millisecond” from the abstract and main text.

8. Additionally, to complement the figures showing the encapsulation process, the author can consider adding plots that monitor the distance between the center of mass of the capsule and the center of mass of the guest with the different steps described in figure 2 highlighted in the plot. In this way, the reader can put together the molecular basis of the self-assembly process with the simulation time.

Response: In past manuscripts and in teaching I used the suggested plots to characterize pathways of protein folding and guest/ligand binding and received comments from both reviewers and graduate students that those plots were too abstract for general readers. So in this revision I increased the number of videos from 10 to 18 in order to reveal as much of the molecular/configurational basis of the self-assembly process as possible.

Reviewers' comments:

Reviewer #2 (Remarks to the Author):

After the revision, the manuscript is a little improved. My concerns are:

(1) The title still seems too general and does not reflect that it is about the host-guest systems.

(2) In the Introduction part, the author should give more examples of the irreversible host-guest complexes and explain their significance.

(3) The priming step is not well defined, why the distance between guest hydrogen atoms and host methylene hydrogen atoms should be greater than 2.6 Å? According to this definition, the length of the priming step seems to depend on the simulation details (for example how many guest molecules is contained in the model system). The physical meaning of the priming step is jeopardized. The author should provide more explanation.

Some further improvement is needed before the paper is accepted.

Reviewer #3 (Remarks to the Author):

In the present version of the manuscript, the author improved some of the concerns pointed out by the reviewers. The manuscript has been reorganized into different sections. The results and the discussion have been extended and contextualized based on some of the suggestions (role of sodium cations, water molecules, and incorporation of new references). Overall, the authors have improved the manuscript based on the referee comments.

Response to the reviewers' comments

I greatly appreciate the precious time and additional comments of Reviewers 2 and 3, which helped further improve the manuscript clarity. Below is my point-to-point response. The changes made according to the comments are colored in red in this submission.

Reviewer 2 (Remarks to the Author):

After the revision, the manuscript is a little improved. My concerns are:

(1) The title still seems too general and does not reflect that it is about the host-guest systems.

Response: In this submission I have accordingly changed the title to “How host and guest self-assemble into a noncovalent irreversible complex and what accelerates the assembly.”

(2) In the Introduction part, the author should give more examples of the irreversible host-guest complexes and explain their significance.

Response: As Reviewer 2 noted above, this manuscript focuses on Cram's host-guest systems of Octacid₄—the first-in-class *noncovalent* irreversible complexes. In Introduction I have referenced aspirin, penicillin, and sotorasib as *covalent* irreversible host-guest complexes and explained their significance. My concern is that providing more examples of irreversible host-guest complexes—namely more *covalent* irreversible host-guest complex examples—could divert readers' attention away from *noncovalent* irreversible complexes.

(3) The priming step is not well defined, why the distance between guest hydrogen atoms and host methylene hydrogen atoms should be greater than 2.6 Å? According to this definition, the length of the priming step seems to depend on the simulation details (for example how many guest molecules is contained in the model system). The physical meaning of the priming step is jeopardized. The author should provide more explanation.

Response: In this submission I have provided the following explanation: “The hydrogen-hydrogen distance cutoff (abbreviated as HH cutoff) was set at 2.6 Å for the following reasons. Visual inspection of all 40 Octacid₄•xylene pathways revealed that xylene was outside of the Octacid₄ cavity as long as the center-of-mass distance cutoff (abbreviated as COM cutoff) for Octacid₄ and its guest was ≥ 7 Å. However, being roughly rectangular in the two-dimensional space, xylene could have its terminal methyl group contact the host cavity portal to slightly enter the host cavity in some of the 40 Octacid₄•xylene pathways at the COM cutoff of 7 or 8 Å. Xylene could also be relatively away from the portal in some pathways at the COM cutoff of ≥ 8 Å. Therefore, rather than using the COM cutoff, the HH cutoff of 2.6 Å was used to avoid both the methyl group contacting the portal (which consequently shortens the ingression time) and the guest being away from the portal (which consequently lengthens the ingression time).”

I have also provided new data in Table S2 to show that the kinetic characterization is independent of the use of the HH cutoff. As apparent from Tables S2, the priming and complexation times for dioxane, xylene and naphthalene determined from the COM cutoff are identical to those determined from the HH cutoff, and the average ingression times from the HH cutoff vs. the COM cutoff are very close (e.g., 95 ps vs 105 ps for xylene).

Reviewer #3 (Remarks to the Author):

In the present version of the manuscript, the author improved some of the concerns pointed out by the reviewers. The manuscript has been reorganized into different sections. The results and the discussion have been extended and contextualized based on some of the suggestions (role of sodium cations, water molecules, and incorporation of new references). Overall, the authors have improved the manuscript based on the referee comments.

Response: I appreciate the positive comments.

REVIEWERS' COMMENTS:

Reviewer #2 (Remarks to the Author):

In the revised paper, the issues raised previously have been addressed. The paper is now suitable for publication.

Response to Reviewer #2's comment

Reviewer #2 (Remarks to the Author):

In the revised paper, the issues raised previously have been addressed. The paper is now suitable for publication.

Response: I appreciate Reviewer #2's precious time and positive comment.